# Manganese-mediated reductive functionalization of activated aliphatic acids and primary amines

Zhan Li[1,3], Ke-Feng Wang[2,3], Xin Zhao [2,3], Huihui Ti[2], Xu-Ge Liu [1✉] & Honggen Wang [1✉]

Alkyl carboxylic acids as well as primary amines are ubiquitous in all facets of biological science, pharmaceutical science, chemical science and materials science. By chemical conversion to redox-active esters (RAE) and Katritzky's N-alkylpyridinium salts, respectively, alkyl carboxylic acids and primary amines serve as ideal starting materials to forge new connections. In this work, a Mn-mediated reductive decarboxylative/deaminative functionalization of activated aliphatic acids and primary amines is disclosed. A series of C-X (X = S, Se, Te, H, P) and C-C bonds are efficiently constructed under simple and mild reaction conditions. The protocol is applicable to the late-stage modification of some structurally complex natural products or drugs. Preliminary mechanistic studies suggest the involvement of radicals in the reaction pathway.

[1] Guangdong Key Laboratory of Chiral Molecule and Drug Discovery, School of Pharmaceutical Sciences, Sun Yat-sen University, Guangzhou 510006, China. [2] Key Laboratory of Molecular Target & Clinical Pharmacology, State Key Laboratory of Respiratory Disease, School of Pharmaceutical Sciences & The Fifth Affiliated Hospital, Guangzhou Medical University, Guangzhou 511436, China. [3]These authors contributed equally: Zhan Li, Ke-Feng Wang, Xin Zhao. ✉email: liuxuge@mail.sysu.edu.cn; wanghg3@mail.sysu.edu.cn

One of the fundamentally important reactions in organic synthesis is the Barbier coupling reaction, which refers to the reductive coupling between an alkyl halide and a carbonyl group in the presence of elemental metal such as zinc, indium, samarium, tin or its salt[1–11]. The metal reduces the alkyl halide in situ via a single electron transfer (SET) mechanism to generate an intermediate organometallic reagent, thereby enabling a one-pot synthesis of secondary or tertiary alcohol in a mild and step-economic fashion (Fig. 1a)[6,7]. In addition to carbonyl compounds, other polar electrophiles including imines, nitriles and α,β-unsaturated carbonyl compounds could also be reacted thanks to the nucleophilic nature of the organometallic reagent generated[8–11].

Alkyl carboxylic acids and alkyl primary amines are widespread in functional molecules[12–15]. Thus, they are ideal starting materials to forge new chemical connections. For this purpose, one feasible way is to activate the carboxylic acids and alkyl primary amines to the corresponding redox-active esters (RAE) and Katritzky's N-alkylpyridinium salts, respectively[16,17]. Previous studies demonstrated that both RAE[18–28] and Katritzky's salts[29–37] are predisposed to accept an electron from low-valent transition metals or organic Lewis bases under photocatalytic reaction conditions, thereby acting as precursors to the corresponding alkyl radicals (Fig. 1b)[18–37]. In connection with the Barbier reaction, it is reasonable to assume that the use of zerovalent metal alone might induce a similar SET process to form an organometallic reagent. If this is the case, the scope of Barbier reaction could be significantly expanded by using ubiquitous alkyl carboxylic acids or primary amines as substrates. In this context, Baran described a reductive Giese reaction of RAEs in the presence of zinc nanopowder[38]. They also realized a RAE-based alkyl Nozaki–Hiyama–Kishi (NHK) reaction with CrCl$_2$/TMSCl[39]. Sun et al. successfully employed gem-difluoroalkenes as electrophile in a zinc-mediated decarboxylative alkenylation of RAEs[40]. Very recently, Larionov developed a decarboxylative phosphine synthesis from RAEs and chlorophosphines with Zinc as stoichiometric reductant and PMDTA (N,N,N',N'',N''-pentamethyldiethylenetriamine) as additive[41]. Despite these exquisite progresses, the development of new method for more types of carbon–heteroatom and carbon–carbon bond formation is still highly desirable. Of note, there is a paucity of deaminative Barbier coupling reactions described in the literature.

We are drawn to the use of elemental manganese as reductant for the decarboxylative and deaminative Barbier type reactions (Fig. 1c). First, manganese is low cost and low toxic[42]. Second, the Lewis acidity of the oxidized Mn$^{2+}$ is much lower than that of Zn$^{2+}$, Sm$^{3+}$, or In$^{3+}$, so that the potential side reactions caused by Lewis acidity could be minimized[43]. And third, the C–Mn bond could serve as a radical surrogate, thus offering a chance for radical coupling reactions[44–46]. It should be noted, however, in the typical Barbier coupling reactions, commercial Mn powder reacts only with the most reactive substrates (allylic halides, α-halogenoesters)[42,47,48]. The less reactive alkyl halides require the use of activated manganese metal, which is often too reactive to be sufficiently chemoselective when complex starting materials are employed[42,49–55].

Herein, we report that the commercial Mn powder is capable to mediate the decarboxylative/deaminative Barbier coupling reactions. The protocol offers a mild access to alkyl radicals via one electron transfer from Mn to the corresponding aliphatic RAEs and Katritzky's N-alkylpyridinium salts. The interception of the alkyl radical leads to diverse carbon–heteroatom and carbon–carbon bond-forming reactions.

## Results

**Initial considerations.** We first investigated decarboxylative/deaminative thiolation reactions. Organosulfur molecules are important motifs found in organic synthesis and functional molecules[56,57]. The most straightforward construction of C

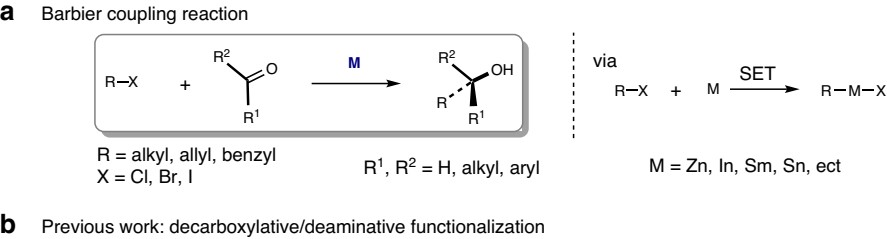

**Fig. 1 Barbier reaction and strategies of decarboxylative/deaminative functionalization. a** Barbier coupling reaction. **b** Previous work: decarboxylative/deaminative functionalization. **c** This work: elemental Mn-mediated reductive decarboxylative/deaminative functionalization.

**Fig. 2 Model reaction of decarboxylative thiolation. a** Decarboxylative thiolation of NHPI ester. **b** Control experiments and the results of using other metal reductant.

(sp³)–S bond is the nucleophilic substitution reaction of an alkyl halide with a mercaptan. While it works well for primary alkyl halides, the use of secondary and tertiary alkyl halides often leads to low yields. Besides, the unpleasant odor of mercaptans may also limits its practical applications. An alternative method to forge C(sp³)–S bond is the reaction of alkyl halides with disulfide reagents under Barbier reaction conditions. Yet, only reactive alkyl halides (e.g., allylic halides, benzylic halides, and α-halogenoesters) were applicable[58–61].

**Decarboxylative thiolation of RAEs.** We initiated our study by focusing on the coupling reaction of piperidine N-hydroxyphthalimide (NHPI) ester **1** with disulfide reagent **2**. Systematic examination of different reaction parameters turned out that with Mn (3.0 equiv) as mediator and 2,2′:6′,2″-terpyridine (50 mol%) as ligand/additive in DMF at 100 °C, the desired coupling product **3** could be formed in 81% yield when disulfide **2** was used as limiting reagent and the loading of **1** was increased to 1.5 equiv (Fig. 2a). A reversed **1**/**2** ratio gave a lower yield of 61% (see Supplementary Table 1). While the use of ultrapure Mn (99.99% purity) gave a similar yield, the omission of the manganese led to no reaction, thus confirming that the manganese is the active mediator. A reaction without terpyridine ligand gave a lower yield of 60%. The use of other commercial metal powders including Fe, Co, Ni, Cu, and Zn in lieu of Mn all led to significantly inferior yields (Fig. 2b). It is known that Zn can reduce RAE by SET. The low yield with zinc in our case is probably due to the undesired reduction of disulfide to thiolate. This was observed experimentally when heating disulfide with zinc or manganese. The former led to the complete decomposition of disulfide to thiol. But the disulfide remained largely untouched in the latter case as detected by GC–MS.

The scope of this decarboxylative thiolation reaction of aliphatic NHPI esters was then explored and found to be quite broad. As shown in Fig. 3, a myriad of diaryl disulfides containing electron-neutral (**3**, **10**, **15**), electron-donating (**11**, **14**, **23**, **26–29**, **36–38**, **42–44**), and electron-withdrawing (**6–9**, **12**, **13**, **16–22**, **24**, **25**, **39–41**) substituents were coupled smoothly to give the corresponding thioethers in moderate to excellent yields. Diheteroaryl disulfide was applicable as well (**30**). Interestingly, the use of dialkyl disulfides in this reaction also delivered the desired products (**31–35**), although the yields are not satisfactory. A notable feature is the survival of halogens (**6–8**, **12**, **16–22**, **24**, **25**, **39–41**) and terminal alkene (**33**). Other important functional groups such as ester (**13**), methoxyl (**11**, **14**, **23**, **26–29**, **36–38**, **42–44**) and trifluoromethoxyl (**9**) were also tolerated. The cross-coupling was successful for various primary, secondary, and tertiary aliphatic NHPI esters, including those containing heterocyclic alkyl groups (**3**, **6–22**, **30–35**), cyclic alkyl group

(**23–26**, **43**, **44**) and acyclic alkyl groups (**27–29**). Primary NHPI ester with α-oxygen substitution was also tolerated and afforded the desired product (**41**). The corresponding carboxylic acids and the decarboxylative hydrogenated alkanes were found to be the major byproducts for the low-yielding cases. Interestingly, the alkyl bromide **1-Br** and iodide **1-I**. also showed good reactivity in this thiolation reaction under the standard reaction conditions. A possible explanation to this observation is that the disulfide may be reduced to a thiolate, and then a $S_N2$ substitution reaction took place to form the thiolation product (see Supplementary Methods)[42,47,48].

The successful development of decarboxylative thiolation prompted us to further explore the feasibility of other chalcogenation reaction via similar approach. Under the above standard reaction conditions, the decarboxylative C–Se and C–Te bond formation reactions proceeded smoothly to afford the desired products (**45–49**). In order to showcase the virtues of our protocol in generating structural diversity for late-stage application, we applied this strategy to different natural products and drug molecules containing a carboxylic acid functional group (**50–53**). Thus, derivatives of pregabalin (**50**), probenecid (**51**), steroids (**52**) and gemfibrozil (**53**) were successfully converted into the desired thiolated products in moderate to good yields. Our protocol thus offers a valuable alternative to the previous light-promoted decarboxlative thiolations by providing broader substrate scope[62].

**Deaminative thiolation of Katritzky's N-alkylpyridinium salts.** The deaminative thiolation of Katritzky's N-alkylpyridinium salts was also investigated. In this case, it was found that the coupling of Katritzky's salt **4** with S-phenyl benzenesulfonothioate reagent **5** (2.5 equiv) in the presence of Mn (5.0 equiv) in DMSO at 70 °C delivered the desired product in an excellent yield of 93% (Fig. 4a, see also Supplementary Table 2). No external additive was needed. Again, the use of ultrapure Mn (99.99% purity) gave an almost identical yield (91%), and no reaction occurred when manganese was omitted (Fig. 4b), confirming elemental Mn played the key role for effectiveness.

The scope and limitation of this deaminative thiolation is shown in Fig. 5. Not surprisingly, a variety of Katritzky's salt and benzenesulfonothioates were well compatible in the reaction, giving the corresponding products in generally good to excellent yields. Of note, S-alkyl benzenesulfonothioates ideally matched the reactivity of Katritzky's salt (**31–35**). As such, a serial of dialkyl thioethers, including the S-glucose thioethers, were constructed in high efficiency. Similarly, under the standard conditions, Se-phenyl benzenesulfonoselenoate was coupled efficiently to afford the selenide in almost quantitative yield (**46**). Also intriguing is the applicability of SS-t-butyl

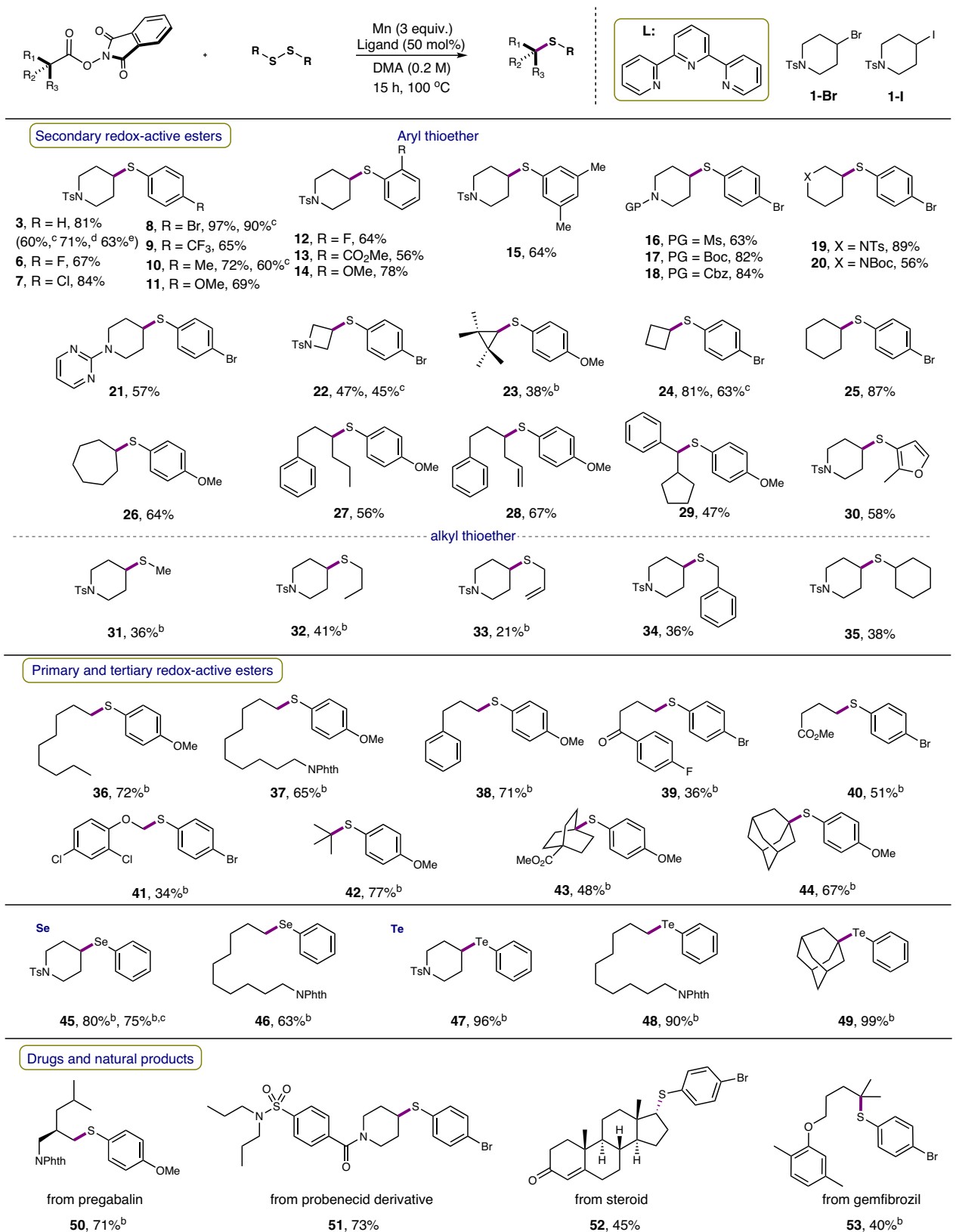

**Fig. 3 Scope of decarboxylative thiolation.** Reaction conditions. [a]NHPI ester (0.3 mmol), disulfide (0.2 mmol), Mn (0.6 mmol), in DMA (0.2 M), 100 °C, N₂, 15 h; yields are for isolated products. [b]NHPI ester (0.4 mmol). [c]Without ligand. [d]**1-Br** was used. [e]**1-I** was used.

**a** Deaminative thiolation

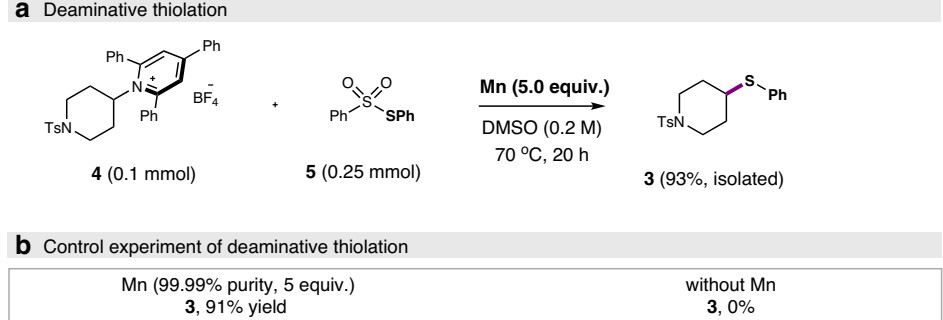

**b** Control experiment of deaminative thiolation

| Mn (99.99% purity, 5 equiv.)<br>**3**, 91% yield | without Mn<br>**3**, 0% |
| --- | --- |

**Fig. 4 Model reaction of deaminative thiolation. a** Deaminative thiolation. **b** Control experiment of deaminative thiolation.

**3**, 93%  **6**, 69%  **7**, 77%  **8**, 84%  **9**, 66%

**11**, 87%  **12**, 81%  **13**, 60%  **14**, 76%  **15**, 91%

**30**, 90%  **54**, 78%  **17**, 78%  **25**, 81%  **55**, 81%

**56**, 74%  **57**, 56%  **31**, 88%  **32**, 70%  **33**, 71%

**34**, 95%  **35**, 69%

thioglycosides

**58**, 67%  **59**, 43%[b]  **60**, 42%  **61**, 39%[b]

**Se**

**46**, 99%

disulfide

**62**, 72%  **63**, 40%  **64**, 56%  **65**, 25%

**Fig. 5 Scope of deaminative thiolation.** Reaction conditions: [a]N-alkylpyridinium salts (0.1 mmol), benzenesulfonothioates (0.25 mmol), Mn (0.5 mmol), in DMSO (0.2 M), 70 °C, N₂, 20 h; yields are for isolated products. [b]Sulfonothioates (0.3 mmol), 80 °C.

**Fig. 6 Scope of decarboxylative hydrogenation.** Reaction conditions: NHPI ester (0.2 mmol), Hantzsch ester (0.4 mmol), Mn (0.6 mmol), in DMF (0.4 M), 50 °C, N$_2$, 16 h; yields are for isolated products.

benzenesulfonothioates in this deaminative Barbier reaction, providing a straightforward route to disulfides products (**62–65**). The low yields for some cases are due to the formation of deaminative hydrogenated alkanes. Our protocol is therefore complementary to the recent light-promoted deaminative thiolations reported by Liao et al.[63].

**Decarboxylative hydrogenation**. The Barton decarboxylation is one of the fundamental reactions in organic synthesis. Yet, the photo- and thermal sensitivity of Barton ester and the use of toxic and odorous tin hydride as reductant represents two major shortcomings. Recently, Baran advanced this chemistry by using stable NHPI ester as a surrogate of Barton ester in the presence of nickel as catalyst and silane and zinc as reductants[64]. Shang found the decarboxylative hydrogenation of NHPI ester could be achieved alternatively by mixing it with Hantzsch ester under photo-irradiation. Amino acid-derived RAEs were unfortunately not compatible[65].

To better define the utility of our manganese-mediated alkyl radical formation reaction, the decarboxylative hydrogenation of NHPI ester was then attempted. We found that, in the presence of Mn (3.0 equiv) and Hantzsch ester (2.0 equiv), the reduction of NHPI esters proceeded smoothly in DMF (0.2 M) at 50 °C, affording the decarboxylative hydrogenation products in generally good yields (see Supplementary Table 3). The protocol was

applicable to a variety of substituted primary, secondary and tertiary RAE (Fig. 6). Those bearing α-heteroatom substitution (**70**, **78**, **83**) were also tolerated well. The derivatives of probenecid (**71**), pregabalin (**82**) and gemfibrozil (**84**) were successfully converted into the desired products as well. The survival of bromo group showcased the high chemoselectivity of this protocol (**80**). The decomposition of the NHPI ester to the corresponding carboxylic acid was found to be the major side reaction pathway.

In addition to the carbon–chalcogen and carbon–hydrogen bond formation reactions, we also explored the feasibility of employing NHPI esters or Katritzky's salts in carbon–carbon bond formation reactions.

**Decarboxylative vinylation**. Transition metal-catalyzed Mizoroki–Heck reaction between olefins and aryl or alkenyl halides to deliver substituted olefins is of great importance in organic synthesis. The extension of this methodology to alkyl halide substrates is typically not trivial due to challenging oxidative addition of low-valent metal and the competing β-hydride elimination, although some progresses have been achieved during the past years. For example, starting from secondary aliphatic NHPI ester, Reisman accomplished an elegant Ni-catalyzed enantioselective decarboxylative coupling with vinyl bromides. Organic tetrakis-(*N*,*N*-dimethylamino)ethylene was used as the

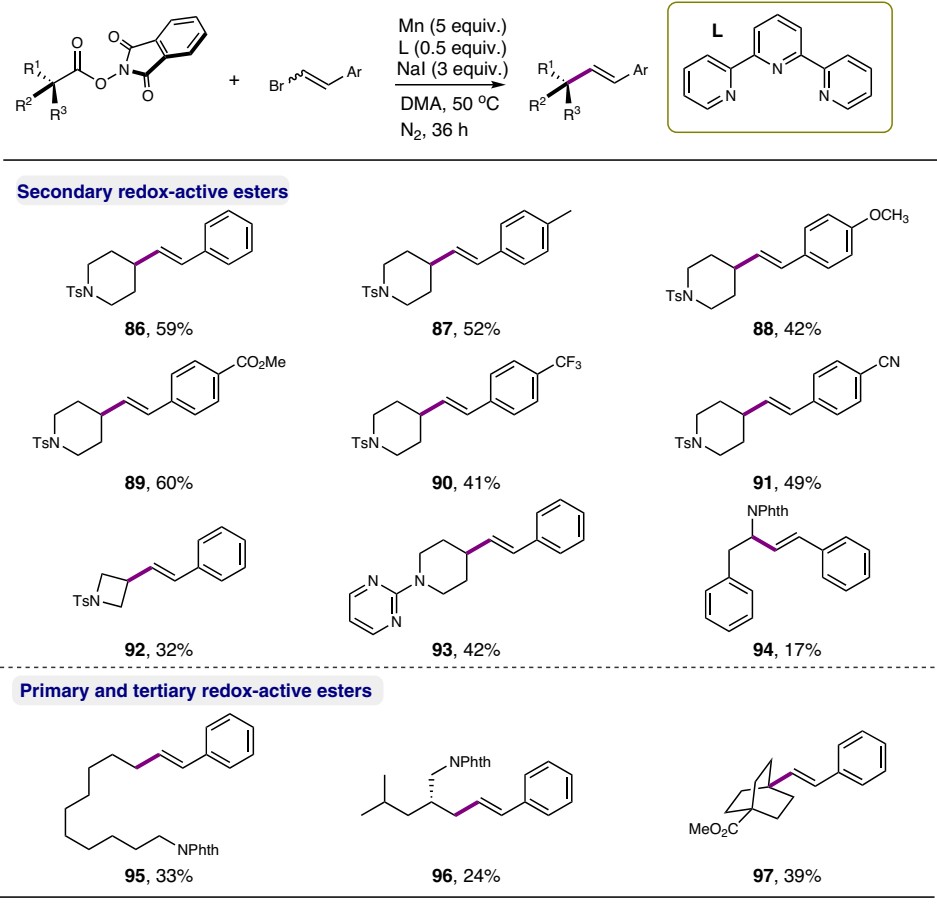

**Fig. 7 Scope of decarboxylative vinylation.** Reaction conditions: NHPI ester (0.2 mmol), bromoethylene (0.6 mmol), 2,2′:6′,2″-terpyridine (0.1 mmol), NaI (0.6 mmol), Mn (1.0 mmol), in DMA (0.4 M), 50 °C, N$_2$, 16 h; yields are for isolated products.

stoichiometric reductant[66]. Overman realized a visible-light photocatalytic coupling of tertiary aliphatic NHPI esters with vinyl bromides by using Hantzsch ester as reductant[67]. Recently, Glorius disclosed a redox-neutral palladium-catalyzed decarboxylative Heck-type coupling of NHPI esters with styrenes under visible-light irradiation[68]. We reasoned the alkyl radical, generated from the single electron-reduction with manganese, might directly add across a vinyl bromide. By a subsequent heteroatom elimination, a C(sp$^3$)–C(sp$^2$) bond formation could be therefore realized.

Indeed, we found that in the presence of Mn (5.0 equiv), NHPI ester (**1**) could react with a $E/Z$ mixture of (2-bromovinyl) benzene (3.0 equiv) to give an alkylated styrene (**86**) in 22% yield as a single $E$ stereoisomer (Fig. 7). The use of NaI (3.0 equiv) and terpyridine (50 mol%) as additives improved the yield to 59% (see Supplementary Table 4). A separate experiment using $Z$-(2-bromovinyl)benzene as coupling partner also delivered the $E$-type product exclusively (54% yield), indicating an oxidative addition of metal to C–Br bond could be excluded and a radical addition/elimination pathway might be operative. Different vinyl bromides featuring varying electron-properties were applicable to the reaction. Primary (**95**, **96**), tertiary (**97**), and α-amino secondary carboxylic acid derivatives (**94**), however, were not good substrates, demonstrating a limitation of this protocol. Again, the formation of the corresponding carboxylic acids and decarboxylative hydrogenated alkanes was responsible for low-to-moderate yields.

**Deaminative difluoroallylation toward the synthesis of *gem*-difluoroalkenes**. *gem*-Difluoroalkenes are intriguing molecules

found broad applications in medicinal chemistry[69,70]. As a carbonyl bioisostere, the introduction of *gem*-difluoroalkene moiety into drug molecules can potentially result in improved pharmaceutical performance. Moreover, *gem*-difluoroalkenes are valuable precursors for the synthesis of a wide variety of fluorine-containing molecules. Previously, the defluorinative S$_N$2′ reaction of 1-(trifluoromethyl)alkenes with different nucleophiles with or without a catalyst offers a straightforward access to substituted *gem*-difluoroalkenes[71]. The electrophilic alkyl halides or aliphatic RAE, were also successfully coupled with 1-(trifluoromethyl) alkenes in the presence of a nickel catalyst and stoichiometric amount of zinc reductant[72,73].

Our above success on the decarboxylative olefination with vinyl bromides hinted that 1-(trifluoromethyl)alkenes might also be suitable coupling partners in the manganese-mediated decarboxylative or deaminative alkylation reactions. While the use of NHPI ester as alkyl source was less fruitful, we did find that the reaction of Katritzky's *N*-alkylpyridinium salts **4** (1.3 equiv) with 1-(trifluoromethyl)alkene (1.0 equiv) in the presence of Mn (5.0 equiv) in DMA at 70 °C delivered the *gem*-difluoroalkenes (**98**) in excellent yield (Fig. 8). Interestingly, no additive was needed at all for reactivity. The scope of this protocol is impressive. 1-(Trifluoromethyl)alkenes with diverse functional groups, such as ether (**99**, **101**, **102**, **107**, **117**), halogen (**100**), ester (**104**) and even unprotected amino group (**109**) could be converted to the desired products successfully (**99–111**). The survival of formyl (**103**) group was surprising, as it is a reactive site in typical Barbier reactions. The reactions of primary alkyl groups were less efficient (**118**, **119**).

**Fig. 8 Scope of deaminative difluoroallylation.** Reaction conditions: *N*-alkylpyridinium salts (0.26 mmol), 1-(trifluoromethyl)alkene (0.2 mmol), Mn (1.0 mmol), in DMA (0.2 M), 70 °C, N$_2$, 20 h; yields are for isolated products.

**Miscellaneous reactions**. Several other reactions were also attempted, and some of preliminary results were shown below. The reaction of NHPI ester **1** with allyl sulfone **120** under the identical reaction conditions to the debrominative olefination successfully provided a reductive allylation product **121** in 42% yield (Fig. 9a). Likewise, the replacement of 1-(trifluoromethyl) alkene with *gem*-difluoroalkene **123** in the defluorinative allylation reaction of Katritzky's salt delivered a monofluoroalkene with good stereoselectivity (30% yield, Fig. 9b). Further, the manganese-mediated decarboxylative and deaminative reaction was also tried in the phosphine synthesis with chlorophosphine as phosphorus source. Without optimization of the reaction conditions, the desired phosphine product could be obtained, but in much less efficiency than Larionov's zinc protocol (Fig. 9c, d)[41].

**Gram-scale synthesis and synthetic applications**. To demonstrate the preparative utility of our method, a one-pot gram-scale decarboxylative thiolation and hydrogenation reactions were conducted. The free carboxylic acids were first reacted with NHPI

in the presence of *N,N*'-diisopropylcarbodiimide and catalytic amount of DMAP. After completion, the volatile was evaporated under vacuum and the residue was subjected to the Mn-mediated thiolation or hydrogenation reaction (Fig. 9e). Pleasingly, the thiolated product **3** and hydrogenated product **85** were both obtained in gram quantity in good yields over two steps. The thioether product could be oxidation with *m*-CPBA to give the corresponding sulfone (**130**) or sulfoxide (**129**) in good yields, depending on the loading of oxidant used (Fig. 9f).

**Mechanistic studies**. It is reasonable to assume that both NHPI esters and Katritzky's salts could accept one electron from elemental manganese. Thereafter, a fragmentation occurs to form a free radical, which might recombine with manganese to form an alkylmanganese species. This species could be regarded as an alkyl radical reservoir constantly releasing alkyl radical out of solvent cage[44–46].

To probe whether the alkyl radical is involved in the reaction mechanism, several radical-clock experiments were conducted.

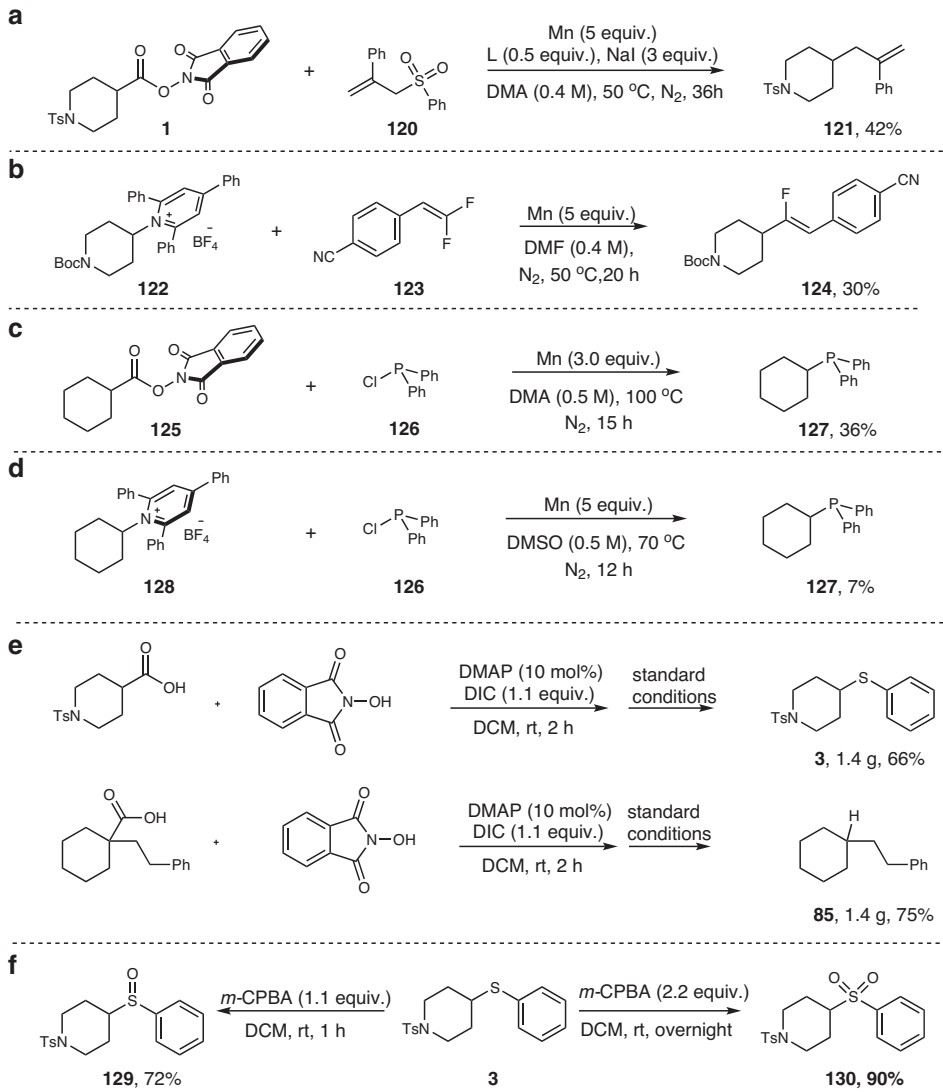

**Fig. 9 Miscellaneous reactions and Synthetic Applications. a** Decarboxylative allylation. **b** Deaminative olefination. **c** Decarboxylative phosphine synthesis. **d** Deaminative phosphine synthesis. **e** Gram-scale reaction (in situ activation protocol). **f** Synthetic application.

The Mn-mediated decarboxylative and deaminative thiolations were chosen for these studies. When the cyclopropyl-substituted NHPI ester **131** was subjected to the reaction, a ring-opening thiolation product **133** was produced exclusively (Fig. 10a). Also, the reaction with a radical ring close precursor resulted in the formation of a mixture of cyclic (**136**) and liner (**137**) products. Radical trapping experiments with TEMPO shut down the reactivity completely, and a TEMPO adduct (**140**) was isolated in 20% yield (Fig. 10b). The same adduct was detected in the hydrogenation and vinylation reaction when TEMPO was introduced (see Supplementary Methods). In all, these results were in good agreement with a radical reaction pathway in the Mn-mediated decarboxylative thiolation reaction. Similar observations found in the deaminative thiolation reactions also pointed to a radial process involved.

The measurements of the reduction potential of the substrates and the sulfur sources suggested the NHPI ester **1** ($E_{1/2} = -1.15$ V vs Ag/AgCl) is more reducing than the disulfide **2** ($E_{1/2} = -1.85$ V vs Ag/AgCl), and the Katritzky's salt **4** ($E_{1/2} = -0.85$ V vs Ag/AgCl) is more reducing than the S-phenyl benzenesulfo-nothioate **5** ($E_{1/2} = -1.35$ V vs Ag/AgCl). Thus, NHPI esters and Katritzky's salts should be reduced first to the corresponding alkyl radicals.

Kinetic studies showed an induction period for both of the decarboxylative and deaminative thiolation reactions (Fig. 10c, d). Interestingly, the addition of different manganese salts at the outset of the reaction could accelerate the reaction to different extents, with $Mn(OAc)_2$ and $MnBr_2$ being the most effective ones for the decarboxylative and deaminative reaction, respectively. The role of manganese salt is not clear. One assumption is that it may act as an electrolyte to facilitate electron transfer in organic medium. Another possibility is that it can activate the NHPI ester or Katritzky's salt, for example, by coordination or π-complexation[43,74,75]. Further mechanistic investigations are needed to better understand the mechanism.

In conclusion, we have developed a Mn-mediated reductive decarboxylative/deaminative functionalization of activated aliphatic acids/amines. A series of C–X (S, Se, Te, H, P) and C–C bonds were efficiently constructed under simple and mild reaction conditions. The protocol was applicable to the late-stage modification of some structurally complex natural products or drugs. Primary mechanistic studies pointed to the involvement of radicals in the reaction pathway. Given the easy availability of the starting materials and the simplicity of the reaction conditions, we anticipate this protocol will find useful applications in organic synthesis.

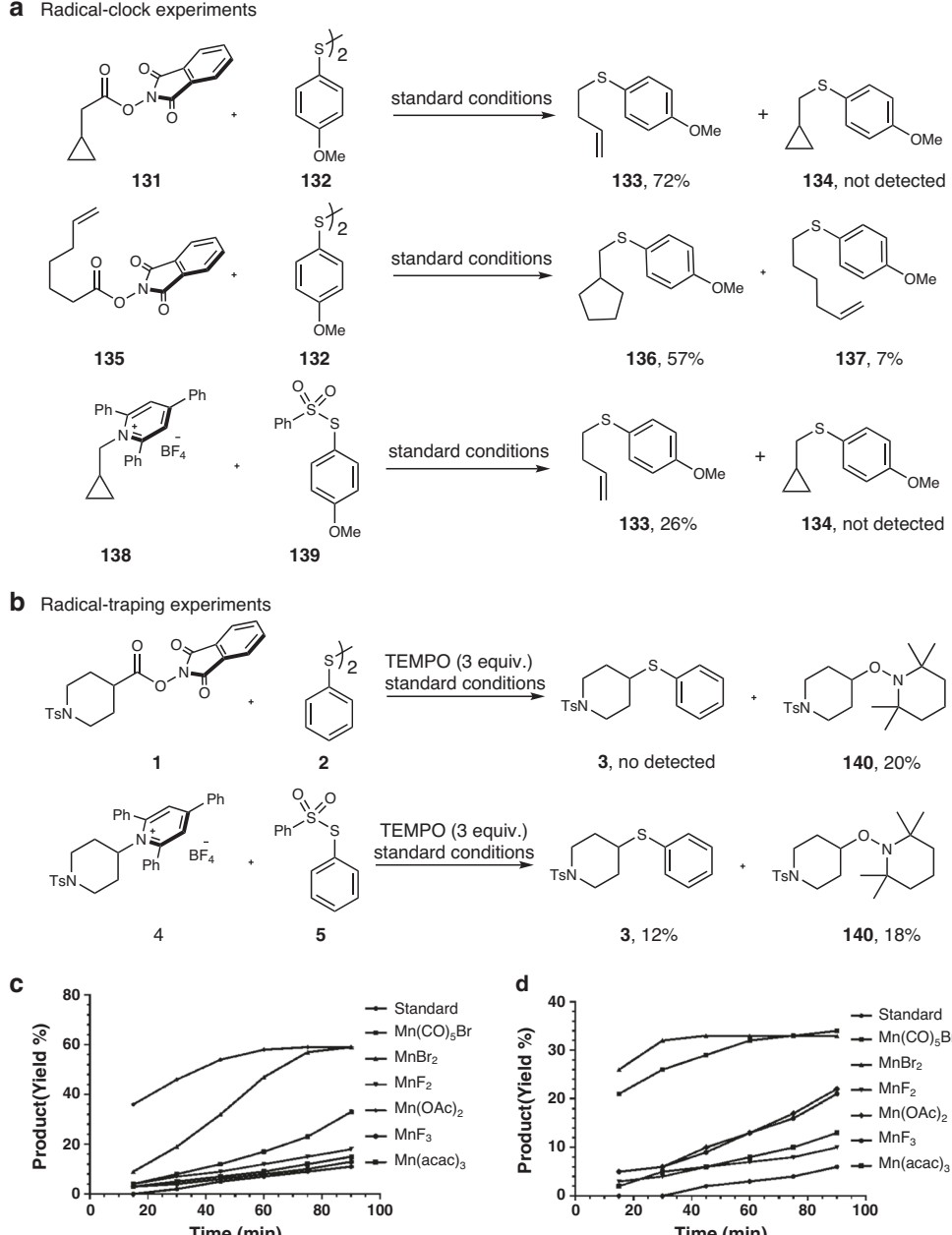

**Fig. 10 Mechanistic studies. a** Radical-clock experiments. **b** Radical-traping experiments. **c** Kinetic analysis of decarboxylative thiolation. **d** Kinetic analysis of deaminative thiolation.

## Methods

**General procedure for decarboxylative thiolation**. Reactions were set up in a $N_2$ filled glove box. To a 10 mL reaction tube equipped with a stirring bar, were added NHPI ester (0.3 or 0.4 mmol, 1.5 or 2 equiv), disulfide (0.2 mmol, 1.0 equiv), Mn (0.6 mmol, 3 equiv) and 2,2′:6′,2″-terpyridine (0.1 mmol, 0.5 equiv), DMA (1.0 mL, 0.2 M) under $N_2$ atmosphere. After that, the resulting mixture was sealed with a screw cap and carried out of the glove box, then the resulting mixture was stirred at 100 °C for 15 h. Then saturated aqueous $NH_4Cl$ was added to quench the reaction. The aqueous layer was then extracted with diethyl ether, and the combined organic layers were washed with brine, dried over anhydrous $Na_2SO_4$, filtered and concentrated. The residue was purified by flash column chromatography on silica gel using Petroleum ether/EtOAc as eluant.

**General procedure for deaminative thiolation**. Reactions were set up in a $N_2$ filled glove box. To a 10 mL reaction tube equipped with a stirring bar, were added pyridinium salts (0.1 mmol, 1.0 equiv), benzensulfonothioates (0.25 mmol, 2.5 equiv), Mn (0.5 mmol, 5 equiv), DMSO (0.5 mL, 0.2 M) under $N_2$ atmosphere. After that, the resulting mixture was sealed with a screw cap and carried out of the glove box, then the resulting mixture was stirred at 70 °C for 20 h. Then saturated

aqueous $NH_4Cl$ was added to quench the reaction. The aqueous layer was then extracted with diethyl ether, and the combined organic layers were washed with brine, dried over anhydrous $Na_2SO_4$, filtered and concentrated. The residue was purified by flash column chromatography on silica gel using Petroleum ether/EtOAc as eluant.

**General procedure for decarboxylative hydrogenation**. Reactions were set up in a $N_2$ filled glove box. To a 10 mL reaction tube equipped with a stirring bar, were added NHPI ester (0.2 mmol), diethyl 1,4-dihydro-2,6-dimethyl-3,5-pyridinedicarboxylate (0.4 mmol, 2.0 equiv), Mn (0.6 mmol, 3 equiv) DMF (0.5 mL, 0.4 M) under $N_2$ atmosphere. After that, the resulting mixture was sealed with a screw cap and carried out of the glove box, then the resulting mixture was stirred at 50 °C for 16 h. Then saturated aqueous $NH_4Cl$ was added to quench the reaction. The aqueous layer was then extracted with diethyl ether, and the combined organic layers were washed with brine, dried over anhydrous $Na_2SO_4$, filtered and concentrated. The residue was purified by flash column chromatography on silica gel using Petroleum ether /EtOAc as eluant.

**General procedure for decarboxylative vinylation**. Reactions were set up in a $N_2$ filled glove box. To a 10 mL reaction tube equipped with a stirring bar, were added NHPI ester (0.2 mmol), vinyl bromide (0.6 mmol, 3.0 equiv), Mn (1.0 mmol, 5.0 equiv), 2,2′:6′,2″-terpyridine (0.1 mmol, 0.5 equiv), DMA (0.5 mL, 0.4 M) under $N_2$ atmosphere. After that, the resulting mixture was sealed with a screw cap and carried out of the glove box, then the resulting mixture was stirred at 50 °C for 36 h. Then saturated aqueous $NH_4Cl$ was added to quench the reaction. The aqueous layer was then extracted with diethyl ether, and the combined organic layers were washed with brine, dried over anhydrous $Na_2SO_4$, filtered and concentrated. The residue was purified by flash column chromatography on silica gel using Petroleum ether/EtOAc as eluant.

**General procedure for deaminative allylation**. Reactions were set up in a $N_2$ filled glove box. To a 10 mL reaction tube equipped with a stirring bar, were added pyridinium salt (0.26 mmol, 1.3 equiv), α-(trifluoromethyl)styrene (0.2 mmol, 1.0 equiv), Mn (1.0 mmol, 5.0 equiv), DMA (1.0 mL, 0.2 M) under $N_2$ atmosphere. After that, the resulting mixture was sealed with a screw cap and carried out of the glove box, then the resulting mixture was stirred at 70 °C for 20 h. Then saturated aqueous $NH_4Cl$ was added to quench the reaction. The aqueous layer was then extracted with diethyl ether, and the combined organic layers were washed with brine, dried over anhydrous $Na_2SO_4$, filtered and concentrated. The residue was purified by flash column chromatography on silica gel using Petroleum ether/EtOAc as eluant.

## Data availability

The authors declare that all the data supporting the findings of this study are available within the article and Supplementary Information files, and also are available from the corresponding author upon reasonable request.

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

## Acknowledgements

This work was supported by the National Natural Science Foundation of China (21801256 and 21971261), the Key Project of Chinese National Programs for Fundamental Research and Development (2016YFA0602900), the Guangdong Basic and Applied Basic Research Foundation (2019A1515011170 and 2020A1515010624), the Fundamental Research Funds for the Central Universities (19ykpy133 and 20ykzd12).

## Author contributions

H.W. and X.-G.L. designed and supervised the project. X.-G.L., Z.L. and K.-F.W. designed and performed the experiments; X.-G.L., X.Z., H.T.Z.L. and K.-F.W. analyzed all the results. H.W. and X.-G.L. prepared the paper. All the authors discussed the results and commented on the paper.

## Competing interests

The authors declare no competing interests.
