## [Peer Review File · Nature Communications]

Reviewers' comments:

Reviewer #1 (Remarks to the Author):

In this manuscript, Liu, Wang and coworkers report several Mn-mediated strategies for the functionalization of redox-active esters and Katritzky N-alkyl pyridinium salts. The authors suggest that these reagents are reduced to the alkyl radical in the presence of superstoichiometric amounts of Mn and proceed to demonstrate radical trapping with various reagents including disulfides and S-phenyl benzenesulfonothioate (C-S bond formation), Hantzsch ester (C-H bond formation), bromoalkenes and trifluoromethylalkenes (C-C bond formation) to name a few.

While the number of examples in the paper is very impressive, the reductive functionalization of these redox-active esters and Katritzky salts is very well precedented in the literature using other mild (and often catalytic) reductants such as photocatalysts and nickel. The authors do a nice job of highlighting this in the manuscript introduction and of highlighting the many related contributions to C-S, C-H and C-C bond formation that have appeared in the literature in the last 5 years. Overall, the results presented in this manuscript seem logical and not unsurprising giving the strong reduction potential of Mn. In fact, related reactions have been reported with Zn (as highlighted in the manuscript intro). It is also not clear what advantage the use of superstoichiometric Mn presents over current state-of-the-art in this field. For these reasons, this report is not novel enough in my opinion to merit publication in Nature Communications and publication as a full paper in a more specialized journal such as Chem. Eur. J. or J. Org. Chem. is recommended.

Minor comments and suggested corrections:

- The manuscript is a little repetitive at times, especially when the "introduction" to each substrate class is presented. For example, the first paragraph of the "Decarboxylative/Deaminative Thiolation" section repeats many of the same concepts that were just stated in the introduction.
- What is the relative reduction potential of the RAE and disulfide in Figure 2? Same question for Figure 3.
- Have the authors tried a reaction with "ligand" in Figure 2? including this data here would be helpful to the reader.
- Overall, the manuscript is very scope focused and could maybe benefit with optimization tables to support the text.
- Generally speaking, literature references are excellent. The discussion of the role of Mn(OAc)₂ and MnBr₂ on the kinetic profile of the reactions could benefit from references to support its potential role as a Lewis acid to facilitate reduction of the RAE or Katritzky salt as others have previously proposed this role.
- Figure 7 should be revised as it is much too small to read currently.

Reviewer #2 (Remarks to the Author):

The manuscript submitted by Liu, Wang and co-workers describes a manganese-mediated reductive decarboxylative/deaminative functionalization of alkyl redox-active esters (NHPI ester) and Katritzky's N-alkylpyridinium salts. In this system, manganese acts as a single electron reductant and no additional catalyst is required. A series of decarboxylative and deaminative C-X (X = S, Se, Te, H, P) and C-C bond formation reactions has been demonstrated under mild conditions with high efficiency, showing a good generality of this Mn-mediated method. Preliminary mechanistic (radical clock, kinetic) studies are consistent with the suggested radical mechanism. Even though several zinc-mediated radical decarboxylative transformations have been reported by Baran and others, the current studies showed the differences between zinc and manganese, as well as its advantages, and further extended this system to the deaminative transformation which has not been realized before with low-valent transition metals. I am therefore supportive of publishing this paper in Nature communication after minor revisions.

- (1) There are several typos in both the manuscript and the supplementary information (e.g., P3: Give; SI, P3 & 1: conditions). And some literature (titles, names) are not following the Nat. Commun. style.
- (2) As mentioned in the third paragraph the potential advantages of using manganese (e.g. commercial Mn powder reacts only with the most reactive substrates of halides), it would be nice to show 1-2 substrates of halide-substituted alkyl acids in Scheme 1 or some comments on this, which would be helpful for others to choose the right conditions for a given reaction.
- (3) Zn is known can reduce redox-active esters by SET, but in this decarboxylative thiolation reaction, only 5% yield was obtained with zinc (Fig. 2). Any explanation? Is PhSSPh the limiting reagent in this transformation? If so, how about the reaction with a reversed RAE/PhSSPh ratio?
- (4) Scheme 1, in the last row, no yield giving for the modification of drugs and natural products.
- (5) Scheme 2, the reaction conditions in footnote a, not with disulfide.
- (6) Larry E. Overman's work (J. Org. Chem. 2015, 80, 6025) on the decarboxylative vinylation with vinyl bromide, representing some earlier examples, should be cited.
- (7) May replace Figure 7 with a high resolution or a bigger one.

Reviewer #3 (Remarks to the Author):

Authors have reported the decarboxylative and deaminative functionalization of alkyl carboxylic acids and amines using metallic manganese. Although large excess of reagents are required to effect the transformation, the strength of the work is the demonstrated vast substrate scope of useful and synthesis of difficult to access organic compounds. Thus, the manuscript can be of interest to synthetic community. Thus, the manuscript can be considered after addressing the following concerns.

- 1) As claimed in title, the terms decarboxylative and deaminative are misleading and should be appropriately modified. Because both acid and amine functionalities are transformed to phthalimido ester and Katritzky's salt in order to effect the desired reactions. Both manoeuvres require stoichiometric reagents causing so much of atomic waste. Actually the ester and ammonium salt that are removed from the substrate not just as carboxylic acid and amines as claimed.
- 2) Schemes are drawn deceptively simple, which can mislead the readers to believe that these reactions are simple to carry out and clean. All the Schemes should reflect the reality, incorporate the byproducts formed and the chemical equations should be balanced.
- 3) What is the role of ligand in the reaction? Why terpyridine act better than other bi and monodentate ligands? Is there in situ complex formations? If so, is the reaction homogeneous or heterogeneous? Further studies to establish the nature of catalysis should be performed.
- 4) Experimental procedures are written in such a way that entire experiments are done inside the glove box. Is it true that even heating was carried out in side the glove box?
- 5) Do the reactions really require strict inert atmosphere? What if the reaction performed without using glove box? Perform few reactions outside the glove box and compare. This is important to find out the usefulness of the method.
- 6) Some spectra including the starting materials show the presence of impurities (for example SI-1-18, and SI-6-2).
- 7) Manuscript needs English language editing
- 8) Throughout the manuscript there are several typos, which require serious attention:
 - Page 2, 3rd paragraph, line 2: Frist should be First
 - Page 3, 1st paragraph, line 7: Give the abundance..
 - Scheme 1, bottom, "feom" instead of 'from'
 - Page 7, 1st paragraph, line 4, silance
 - Page 8, 1st paragraph, line 2, to give the an alkylated... and last line "not compatible well"
 - Figure 5a caption: in suit activation.

Figure 5a, second scheme: provide straight forward drawing rather than just twisting, which can confuse the readers: maintain similarity in starting material and product.

Page 11, 1st paragraph, line 7: it can active.

9) As concluded this is not a simple reaction conditions. Use of 3 to 5 equivalents of Mn can become very much caveat on application of this protocol.

10) Why the reactions require such excess amount of metal? Can't be reduced the required Mn quantity through optimisation studies?

11) References are not formatted as per journal guidelines.

Response to the reviewers

Reviewers' comments:

Reviewer #1 (Remarks to the Author):

In this manuscript, Liu, Wang and coworkers report several Mn-mediated strategies for the functionalization of redox-active esters and Katritzky N-alkyl pyridinium salts. The authors suggest that these reagents are reduced to the alkyl radical in the presence of superstoichiometric amounts of Mn and proceed to demonstrate radical trapping with various reagents including disulfides and S-phenyl benzenesulfonothioate (C–S bond formation), Hantzsch ester (C–H bond formation), bromoalkenes and trifluoromethylalkenes (C–C bond formation) to name a few.

While the number of examples in the paper is very impressive, the reductive functionalization of these redox-active esters and Katritzky salts is very well precedented in the literature using other mild (and often catalytic) reductants such as photocatalysts and nickel. The authors do a nice job of highlighting this in the manuscript introduction and of highlighting the many related contributions to C–S, C–H and C–C bond formation that have appeared in the literature in the last 5 years. Overall, the results presented in this manuscript seem logical and not unsurprising giving the strong reduction potential of Mn. In fact, related reactions have been reported with Zn (as highlighted in the manuscript intro). It is also not clear what advantage the use of superstoichiometric Mn presents over current state-of-the-art in this field. For these reasons, this report is not novel enough in my opinion to merit publication in Nature Communications and publication as a full paper in a more specialized journal such as Chem. Eur. J. or J. Org. Chem. is recommended.

Minor comments and suggested corrections:

The manuscript is a little repetitive at times, especially when the "introduction" to each substrate class is presented. For example, the first paragraph of the "Decarboxylative/Deaminative Thiolation" section repeats many of the same concepts that were just stated in the introduction.

Response:

The repetitive introduction of decarboxylation/deamination concept was removed as suggested.

What is the relative reduction potential of the RAE and disulfide in Figure 2? Same question for Figure 3.

Response:

Thanks for this good suggestion. We have measured the reduction potentials of the corresponding substrates. As expected, RAE **1** ($E_{1/2} = -1.15$ V vs Ag/AgCl) is more reducing than the disulfide **2** ($E_{1/2} = -1.85$ V vs Ag/AgCl), and the Katritzky's salt **4** ($E_{1/2} = -0.85$ V vs Ag/AgCl) is more reducing

than the S-phenyl benzenesulfonothioate **5** ($E_{1/2} = -1.35$ V vs Ag/AgCl). These results are in good agreement with our mechanistic hypothesis that RAEs and Katritzky's salts are reduced first to the corresponding alkyl radicals.

These results were commented in the main text.

Have the authors tried a reaction with "ligand" in Figure 2? including this data here would be helpful to the reader.

Response: The reaction outcomes (using other metal reductants) presented in Figure 2 were corresponding to the ones with ligand terpyridine used. The figure is modified to better clarify the results.

Overall, the manuscript is very scope focused and could maybe benefit with optimization tables to support the text.

Response: The detailed optimization tables were included in the supporting information.

Generally speaking, literature references are excellent. The discussion of the role of $Mn(OAc)_2$ and $MnBr_2$ on the kinetic profile of the reactions could benefit from references to support its potential role as a Lewis acid to facilitate reduction of the RAE or Katritzky salt as others have previously proposed this role.

Response: Two new references regarding the use of manganese salt as Lewis acid were cited: *Synlett*, **2004**, 5, 0846–0850; 2) *Tetrahedron Lett.* **1991**, 32, 5817-5820.

Figure 7 should be revised as it is much too small to read currently.

Response: A new figure with higher resolution was provided.

Reviewer #2 (Remarks to the Author):

The manuscript submitted by Liu, Wang and co-workers describes a manganese-mediated reductive decarboxylative/deaminative functionalization of alkyl redox-active esters (NHPI ester) and Katritzky's N-alkylpyridinium salts. In this system, manganese acts as a single electron reductant and no additional catalyst is required. A series of decarboxylative and deaminative C-X (X = S, Se, Te, H, P) and C-C bond formation reactions has been demonstrated under mild conditions with high efficiency, showing a good generality of this Mn-mediated method. Preliminary mechanistic (radical clock, kinetic) studies are consistent with the suggested radical mechanism. Even though several zinc-mediated radical decarboxylative transformations have been reported by Baran and others, the current studies showed the differences between zinc and manganese, as well as its advantages, and further extended this system to the deaminative transformation which has not been realized before

with low-valent transition metals. I am therefore supportive of publishing this paper in Nature communication after minor revisions.

(1) There are several typos in both the manuscript and the supplementary information (e.g., P3: Give; SI, P3 & 1: conditions). And some literature (titles, names) are not following the Nat. Commun. style.

Response: The typos were carefully checked and revised. The style of literature is modified according to *Nat. Commun.* style.

(2) As mentioned in the third paragraph the potential advantages of using manganese (e.g. commercial Mn powder reacts only with the most reactive substrates of halides), it would be nice to show 1-2 substrates of halide-substituted alkyl acids in Scheme 1 or some comments on this, which would be helpful for others to choose the right conditions for a given reaction.

Response: we have prepared two analogs of NHPI ester **1**: bromide **1-Br** and iodide **1-I**. Interestingly, both substrates showed good reactivity in this thiolation reaction under the standard reaction conditions. An explanation to this observation is that the disulfide may be reduced to a thiolate, and then a S_N2 substitution reaction took place to form the thiolation product. Separate experiments showed that the reaction of **1-Br** or **1-I** with thiolate gave a moderate yield of the thiolated product, confirming our hypothesis.

The results were added to Scheme 1 and commented in the main text.

(3) Zn is known can reduce redox-active esters by SET, but in this decarboxylative thiolation reaction, only 5% yield was obtained with zinc (Fig. 2). Any explanation? Is PhSSPh the limiting reagent in this transformation? If so, how about the reaction with a reversed RAE/PhSSPh ratio?

Response: a) The low yield when zinc was used is probably due to the undesired reduction of disulfide to the corresponding thiolate. This was observed experimentally when heating disulfide with zinc or manganese. The reaction outcomes were determined by GC-MS. The former (zinc) led to the complete decomposition of disulfide to thiol. But the disulfide remained largely untouched in the latter case (manganese). Another possibility is that the Lewis acidity of the Zn^{2+} formed in situ may lead to the decomposition of the starting materials. Indeed, when $ZnCl_2$ was added to the manganese reaction, only 51% yield was obtained.

b) PhSSPh was used as the limiting reagent in our transformation. A reversed RAE/PhSSPh ratio gave a lower yield of 61%.

These results were commented in the main text.

(4) Scheme 1, in the last row, no yield giving for the modification of drugs and natural products.

Response: The yields are now given.

(5) Scheme 2, the reaction conditions in footnote a, not with disulfide.

Response: “disulfide” was corrected as “benzenesulfonothioates”.

(6) Larry E. Overman’s work (J. Org. Chem. 2015, 80, 6025) on the decarboxylative vinylation with

vinyl bromide, representing some earlier examples, should be cited.

Response: This reference is now cited and commented in the main text.

(7) May replace Figure 7 with a high resolution or a bigger one.

Response: A new figure with higher resolution was provided.

Reviewer #3 (Remarks to the Author):

Authors have reported the decarboxylative and deaminative functionalization of alkyl carboxylic acids and amines using metallic manganese. Although large excess of reagents are required to effect the transformation, the strength of the work is the demonstrated vast substrate scope of useful and synthesis of difficult to access organic compounds. Thus, the manuscript can be of interest to synthetic community. Thus, the manuscript can be considered after addressing the following concerns.

1) As claimed in title, the terms decarboxylative and deaminative are misleading and should be appropriately modified. Because both acid and amine functionalities are transformed to phthalimido ester and Katritzky's salt in order to effect the desired reactions. Both manoeuvres require stoichiometric reagents causing so much of atomic waste. Actually the ester and ammonium salt that are removed from the substrate not just as carboxylic acid and amines as claimed.

Response: We agree with this statement. The terms “decarboxylative and deaminative” are removed from the title.

2) Schemes are drawn deceptively simple, which can mislead the readers to believe that these reactions are simple to carry out and clean. All the Schemes should reflect the reality, incorporate the byproducts formed and the chemical equations should be balanced.

Response: For the reaction of NHPI esters, the major byproducts were determined to be the corresponding carboxylic acids and decarboxylative hydrogenated alkanes. For the reaction of Katritzky's salt, the major byproduct is the deaminative hydrogenated alkane. These results were commented in the main text. It might not be that interesting to give the exact amount (yield) of each byproduct in the schemes.

3) What is the role of ligand in the reaction? Why terpyridine act better than other bi and monodentate ligands? Is there in situ complex formations? If so, is the reaction homogeneous or heterogeneous? Further studies to establish the nature of catalysis should be performed.

Response: The following reactions were conducted to determine the role of terpyridine ligand:

1) The use of chiral tridentate ligand

Three chiral tridentate ligands in lieu of terpyridine were used in the reaction. No apparent ee was found. These results indicate that the ligand or Mn-ligand complex is not involved in the C-S bond formation step.

2) The use of pre-formed Mn-terpyridine complex as additive

Mn(terpy)Cl₂ complex, a tentative mediator/catalyst for the reaction, was prepared according to a precedent literature (*Angew. Chem. Int. Ed.* **2016**, 55, 14369-14372). Its application (0.5 equiv or 0.1 equiv.) in the thiolation reaction give an inferior yield as compared to terpyridine.

3) Kinetics

The kinetics for both the reactions with terpyridine and without terpyridine were measured. Out of our expectation, a slow reaction rate was observed when terpyridine was used.

4) Some results without ligand

We have tried more examples without terpyridine ligand. The results were shown below. We can see the role of terpyridine is not decisive. For compounds **8**, **22** and **45**, similar yields were observed.

In all, the use of terpyridine in our reaction is beneficial for the yield. However, its role is not decisive for both the reactivity and efficacy. Similar observation was actually found in Hu's manganese-mediated reductive transamidation reaction wherein bipyridine was used as ligand (*J. Am. Chem. Soc.* **2018**, 140, 6789–6792). The above results gave no clues on the tricky role of terpyridine in our reaction.

The discussion on the role of terpyridine was included in SI.

4) Experimental procedures are written in such a way that entire experiments are done inside the glove box. Is it true that even heating was carried out inside the glove box?

Response: We have correct this section. The reaction mixture was actually heated outside the glove

box.

5) Do the reactions really require strict inert atmosphere? What if the reaction performed without using glove box? Perform few reactions outside the glove box and compare. This is important to find out the usefulness of the method.

Response: As suggested, we have performed the reaction on bench. While the reaction without N₂ protection gave no desired product, the reaction under N₂ atmosphere gave a yield of 76%, slightly lower than the standard reaction conditions (81%).

6) Some spectra including the starting materials show the presence of impurities (for example SI-1-18, and SI-6-2).

Response: The impurities were removed and the spectras were re-recorded.

7) Manuscript needs English language editing

Response: The language was corrected by colleagues with good English language skills.

8) Throughout the manuscript there are several typos, which require serious attention:

Page 2, 3rd paragraph, line 2: Frist should be First

Page 3, 1st paragraph, line 7: Give the abundance..

Scheme 1, bottom, “feom” instead of ‘from’

Page 7, 1st paragraph, line 4, silance

Page 8, 1st paragraph, line 2, to give the an alkylated... and last line “not compatible well”

Figure 5a caption: in suit activation.

Figure 5a, second scheme: provide straight forward drawing rather than just twisting, which can confuse the readers: maintain similarity in starting material and product.

Page 11, 1st paragraph, line 7: it can active.

Response: We have checked the manuscript carefully, and the corresponding corrections were made.

9) As concluded this is not a simple reaction conditions. Use of 3 to 5 equivalents of Mn can become very much caveat on application of this protocol.

Response: We agree that the use of excess amount of Mn is a drawback in this reaction. But this is a common challenge when elemental manganese is used as the reductant. For examples, in the field of transition metal-catalyzed reductive cross-coupling reactions, Reisman (*J. Am. Chem. Soc.* 2013, 135, 7442-7445.; *J. Am. Chem. Soc.* 2014, 136, 14365-14368; *J. Am. Chem. Soc.* 2015, 137, 10480-10483; *J. Am. Chem. Soc.* 2017, 139, 5684-5687; *J. Am. Chem. Soc.* 2018, 140, 139-142), Shenvi (*J. Am. Chem. Soc.* 2018, 140, 11317-11324) used 3 to 5 equivalents of Mn as reductant. In a recent nickel-catalyzed reductive C–S bond formation reaction (*Org. Chem. Front.*, 2017, 4, 31–36), 5 equivalents of Mn was used. And in Hu’s reductive transamidation reaction, up to 10 equivalents of Mn was used (*J. Am. Chem. Soc.* 2018, 140, 6789-6792).

Some closely related examples. Weix used 3 equivalents of Mn in the decarboxylative alkylation of RAEs (*Angew. Chem. Int. Ed.* 2017, 56, 11901–11905), and Watson also used 3 equivalents of Mn in his deaminative reductive cross-couplings (*Org. Lett.* 2019, 21, 2941-2946).

10) Why the reactions require such excess amount of metal? Can’t be reduced the required Mn quantity through optimisation studies?

Response: As responded above, it is a common challenge to reduce the loading of elemental manganese in reduction reactions. We have tried different solvents and additives, but no promising solution to this problem is obtained.

11) References are not formatted as per journal guidelines.

Response: the format has been corrected.

REVIEWER COMMENTS

Reviewer #2 (Remarks to the Author):

This work deals with the use of elemental manganese as reductant for decarboxylative and deaminative transformations. As demonstrated, several reactions can be well established with activated aliphatic acids and primary amines, with an impressive large number of examples. The difference between Mn and other metals (e.g. Zn, Fe, etc.) as mentioned by the authors in the third paragraph was a good starting point for this research.

Lines 33-36: "It should be noted, however, in the typical Barbier coupling reactions, commercial Mn powder reacts only with the most reactive substrates (allylic halides, halogenoesters. The less reactive alkyl halides require the use of activated manganese metal, which is often too reactive to be sufficiently chemo selective when complex starting materials are employed."

- Therefore, how about halide (Cl, Br, I)-containing acid or amine substrates (e.g. 6-bromohexanoic acid)? Can halide and redox-active ester be well distinguished under these Mn-mediated reaction conditions (thiolation and also other reaction types)? A comparison study with other metals may further manifest the advantages of manganese.

A reversed RAE/PhSSPh ratio (with RAE as the limiting component) gave a lower yield of 61%. Any explanation?

Reviewer #3 (Remarks to the Author):

Authors have addressed the points raised by the Reviewers, performed additional experiments and provided further explanations both in manuscript and SI.

Authors should be judicious in use of reagents. Using 50 mol % of expensive ligand for few extra % of product yield is not justified. In some cases, the product yield is poor even after the use of such large amount of ligand.

After going over the authors rebuttal and revised manuscript, I recommend the manuscript for publication in Nature Chemistry.

Reviewer #2 (Remarks to the Author):

This work deals with the use of elemental manganese as reductant for decarboxylative and deaminative transformations. As demonstrated, several reactions can be well established with activated aliphatic acids and primary amines, with an impressive large number of examples. The difference between Mn and other metals (e.g. Zn, Fe, etc.) as mentioned by the authors in the third paragraph was a good starting point for this research.

Lines 33-36: "It should be noted, however, in the typical Barbier coupling reactions, commercial Mn powder reacts only with the most reactive substrates (allylic halides, halogenoesters. The less reactive alkyl halides require the use of activated manganese metal, which is often too reactive to be sufficiently chemo selective when complex starting materials are employed." Therefore, how about halide (Cl, Br, I)-containing acid or amine substrates (e.g. 6-bromohexanoic acid)? Can halide and redox-active ester be well distinguished under these Mn-mediated reaction conditions (thiolation and also other reaction types)? A comparison study with other metals may further manifest the advantages of manganese.

Response:

- (a) We have prepared two halogen-containing (Br, Cl) NHPI esters. The reaction of both substrates showed that the thiolation on both the halogen and NHPI ester sites were observed (eq a and b). Nevertheless, under the decarboxylative hydrogenation conditions, chemoselective hydrogenation of the NHPI ester was detected (eq c).

A possible explanation to this is that, for thiolation reaction, disulfide may be reduced to a thiolate, and then a $\text{S}_{\text{N}}2$ substitution reaction took place to form the thiolation product. Actually, the reactivity of alkyl halide **1-Br** and **1-I** was commented previously in the main-text and the results were previously included in the SI.

(2) We have difficulties to prepare the halogen-containing Katritzky's *N*-alkylpyridinium salts after many tries, since their precursors- the halogen-containing amines are prone to undergo intramolecular or intermolecular SN2 displacement. This also hinted that the halogen-containing Katritzky's *N*-alkylpyridinium salts should represent a very minor class of substrates which may be used in synthesis.

A reversed RAE/PhSSPh ratio (with RAE as the limiting component) gave a lower yield of 61%. Any explanation?

Response:

This is due to the undesired side reactions of RAEs, for examples, decomposition to carboxylic acids and decarboxylative hydrogenations, which are typically observed for many cases, especially for the low-yielding substrates.

REVIEWERS' COMMENTS:

Reviewer #2 (Remarks to the Author):

Authors have addressed my concerns, and also included the results in the revised manuscript.